# Trading Personalization for Accuracy:
# Data Debugging in Collaborative Filtering

**Long Chen**[1]**, Yuan Yao**[1]**, Feng Xu**[1]**, Miao Xu**[2,3]**, Hanghang Tong**[4]

[1]State Key Laboratory for Novel Software Technology, Nanjing University, China
[2]The University of Queensland, Australia    [3]RIKEN AIP, Japan
[4]University of Illinois Urbana-Champaign, USA
ronchen@smail.nju.edu.cn, {y.yao, xf}@nju.edu.cn, miao.xu@riken.jp, htong@illinois.edu

## Abstract

Collaborative filtering has been widely used in recommender systems. Existing
work has primarily focused on improving the prediction accuracy mainly via either
building refined models or incorporating additional side information, yet has largely
ignored the inherent distribution of the input rating data. In this paper, we propose a
data debugging framework to identify overly personalized ratings whose existence
degrades the performance of a given collaborative filtering model. The key idea
of the proposed approach is to search for a small set of ratings whose editing
(e.g., modification or deletion) would near-optimally improve the recommendation
accuracy of a validation set. Experimental results demonstrate that the proposed
approach can significantly improve the recommendation accuracy. Furthermore, we
observe that the identified ratings significantly deviate from the average ratings of
the corresponding items, and the proposed approach tends to modify them towards
the average. This result sheds light on the design of future recommender systems
in terms of balancing between the overall accuracy and personalization.

## 1   Introduction

Recommender systems have been widely used in many real-world applications to suggest items
to users. As a key building block of recommender systems, collaborative filtering aims to model
users' preferences to items based on the existing feedback/ratings. Classic collaborative filtering
methods include nearest-neighbor methods [14], factorization-based methods [4], deep learning based
methods [19, 3], etc.

In literature, many efforts have been spent on improving the prediction accuracy of collaborative
filtering [16]. The main focus of the existing work can be divided into two categories. The first
category aims to refine the objective function so as to better model the input ratings (e.g., [13, 28,
3, 15]), and the second aims to incorporate additional inputs such as social relationships or item
attributes (e.g., [20, 25, 12, 9, 24]). However, little attention is paid to the inherent data distribution
of the input ratings. That is, although collaborative filtering benefits from similar interests among
similar users, overfitting to overly personalized ratings from certain users may deteriorate, instead
of help, the overall recommendation. For example, suppose a set of users share most of their movie
interests (e.g. comedy, romance), and one user favors extreme-horror films. Although such ratings
might be important to reflect this specific user's personal preference, they could downgrade the
recommendation quality to the vast majority of users (i.e., recommending unwanted extreme-horror
films to these users). This motivates us to ask: *are all ratings helpful in collaborative filtering*,
and if not, *how can we mitigate harmful (i.e., overly personalized) ones to improve the overall
recommendation accuracy?*

The questions we ask in this paper are related to, but bear subtle differences from *anomaly detection*, *robust learning*, and *adversarial attacking*. First, anomaly detection in collaborative filtering typically assumes that there is inconsistency among a particular user's ratings, and clustering-based methods have been developed to detect such inconsistency [10, 18, 23, 17, 1]; however, it is not clear to what extent the detected anomalies hurt the overall recommendation accuracy. Second, robust learning seeks to stably learn the models even when there exist noises [26, 8] or bias [21, 22] in the training data; however, these methods do not explicitly identify the noises/bias, and the relationship between such noises/bias and over-personalization is unclear. Finally, adversarial attacking [6, 7, 2] targets to degrade the recommender system by manipulating the existing user accounts or ratings; in contrast, we aim to improve the system by identifying and editing overly personalized ratings.

In this paper, we aim to answer the above questions from a data debugging perspective. Analogizing to software debugging, we propose an algorithm CFDEBUG to identify and edit overly personalized ratings (referred to as *maverick ratings* in this paper), so as to improve the accuracy of factorization-based collaborative filtering. Specially, we formalize an optimization objective whose minimization can simultaneously learn the recommendation model and identify maverick ratings. Due to the difficulty of solving the objective and learning the discrete ratings, we do alternative optimization and follow the idea of [11, 7], conducting a gradient descent on the rating matrix and treating the largest-updated entries as maverick ones. To avoid overfitting, we learn the recommendation model on training data, and update the rating matrix on validating data. By splitting the original data into multiple training/validating combinations, we finally ensemble the learned results. Experimental results demonstrate that the proposed approach can significantly improve the recommendation accuracy by identifying and editing some existing ratings. By analyzing the identified ratings, we observed that they significantly deviate from the average ratings of the corresponding items, and the proposed approach tends to modify them towards the average.[1] Our findings may shed light on the design of future recommender systems.

The rest of the paper is organized as follows. Section 2 provides the problem statement. Section 3 describes the proposed approach, and Section 4 shows the experimental results. Section 5 concludes.

## 2   Preliminaries and Problem Statement

*Preliminaries: Factorization-based CF*. Given a rating matrix $A \in \mathbb{R}^{m \times n}$ of $m$ users and $n$ items as input, where each entry $a_{i,j}$ represents the observed rating from user $i$ to item $j$, the goal of collaborative filtering (CF) is to predict a user's rating to a target item that is unobserved in $A$. To solve the problem, a common assumption is to approximate $A$ with a low-rank structure. For example, by assuming rank $k$ (where $k \ll min(m,n)$) of the input rating matrix, we can factorize it into two low-rank matrices $U$ and $V$, i.e.,

$$L = \min_{U,V} \sum_{(i,j) \in \Omega} (a_{i,j} - u_i v_j^T)^2 + \lambda_u \|u_i\|^2 + \lambda_v \|v_j\|^2, \tag{1}$$

where $U \in \mathbb{R}^{m \times k}$ and $V \in \mathbb{R}^{n \times k}$ are the parameters to learn, $\Omega$ represents the indices of the observed ratings in $A$, $u_i \in \mathbb{R}^k$ is the $i$-th row of $U$, $v_j \in \mathbb{R}^k$ is the $j$-th row of $V$, and $\lambda_u$ and $\lambda_v$ are regularization parameters to avoid overfitting. The aim is to make $a_{i,j} \simeq u_i v_j^T$, $\forall (i,j) \in \Omega$. With $U$ and $V$ learned from the above formulation, we can obtain the unobserved entries of matrix $A$ as $\hat{a}_{i,j} = u_i v_j^T$.

*Problem Statement*. In this work, we aim to study the trade-off between personalization and overall accuracy in CF from a data debugging perspective. We use $\Omega$ and $\Phi$ to denote the observed ratings and the maverick ratings (i.e., $\Phi \subset \Omega$). The induced adjacency matrix of $\Phi$ is denoted as $C \in \mathbb{R}^{m \times n}$, whose entry indicates whether (or to what extent) it is a maverick rating. We further use $\Theta(\cdot)$ to denote the learned parameters from the CF model (e.g., $U$ and $V$ in Eq (1)), and $L(\Theta)$ to denote the loss computed based on the parameters $\Theta$. With the above notations, we define the data debugging problem for CF as follows. **Given** (1) an $m \times n$ rating matrix $A$ whose observed ratings are included in set $\Omega$, and (2) a specific collaborative filtering model depicted by the loss function $L$; the goal is to **find** a set of maverick ratings $\Phi$ in $\Omega$ whose editing (e.g., modification, deletion) could improve the recommendation accuracy of the collaborative filtering model $L$.

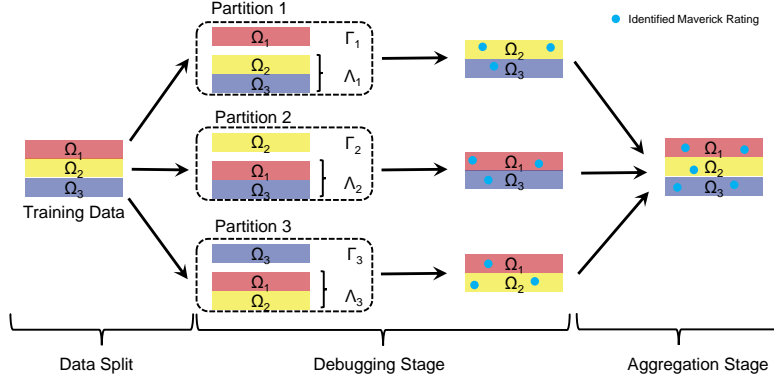

Figure 1: The overview of the proposed CFDEBUG. $\Lambda_i$ and $\Gamma_i$ are training and validation sets, respectively.

# 3 The Data Debugging Approach

In this section, we present the proposed data debugging approach CFDEBUG. Fig. 1 shows the overall framework of the proposed approach. There are two stages of CFDEBUG. In the first debugging stage, we randomly divide the existing ratings into several folds (i.e., $\Omega_1$, $\Omega_2$, and $\Omega_3$ in the figure) and follow the way similar to cross-validation to search for the maverick ratings. That is, each fold is used once as the *validation set* to debug the rest folds. For example, in Partition 1, fold $\Omega_1$ is used as the validation set (i.e., $\Gamma_1$), and the rest two folds are *debug set* (i.e., $\Lambda_1$ with $\Gamma_1 + \Lambda_1 = \Omega$) from which the maverick ratings are identified. In the second aggregation stage, we aggregate the debugging results for each fold to identify the final maverick ratings. For example, in the figure, $\Omega_1$ has been debugged twice (when $\Omega_2$ and $\Omega_3$ are used as validation sets, respectively), and the two debugging results are combined to form the final results.

## 3.1 The Debugging Stage

In the following, we take one partition as an example to explain the key idea of the debugging stage. For brevity, we omit the subscript in the following. For example, we use $\Gamma$ and $\Lambda$ to denote the validation set and the debug set, respectively. Then, we design the optimization framework for each partition as follows,

$$\Phi = \underset{\Phi \subset \Lambda}{\arg\min}\, L_\Gamma(\Theta(\Lambda - \Phi)), \quad s.t. \quad |\Phi| \le K, \tag{2}$$

where we aim to identify the maverick rating set $\Phi$, and we constrain its maximum size to $K$. As we can see from the above equation, the key idea of CFDEBUG is to spot the set $\Phi$, and if we further edit (e.g., delete, modify) the ratings in the set $\Phi$, it would mostly decrease the loss function $L_\Gamma$ defined on the validation set $\Gamma$.

To further illustrate the above optimization framework, we take the CF model in Eq. (1) as an example, where $\Theta = \{\widetilde{U}, \widetilde{V}\}$ contains the parameters learned using $(\Lambda - \Phi)$ as input and $L(\Theta)$ denotes the squared loss. In other words, the model parameters $\Theta = \{\widetilde{U}, \widetilde{V}\}$ are obtained from the training set $(\Lambda - \Phi)$ and the loss function is measured on the validation set $\Gamma$. We have

$$L_\Gamma(\Theta) = \sum_{(i,j) \in \Gamma} (a_{i,j} - \widetilde{u}_i \widetilde{v}_j^T)^2, \tag{3}$$

where $\widetilde{u}_i$ denotes the $i$-th row of $\widetilde{U}$ and $\widetilde{v}_j$ denotes the $j$-th row of $\widetilde{V}$, and we ignore the regularization terms in Eq. (1) for brevity. Then, the goal becomes to find the set $\Phi$ that minimizes the above equation.

*(A) Solving Eq. (2).* Solving Eq. (2) is challenging due to the following reasons. First, directly searching the solution space is computationally intractable. For example, if we aim to find top-$K$ maverick ratings from the existing ratings, the search space is $O(\binom{|\Lambda|}{K})$ which is combinatorial. Moreover, the existing ratings are usually from a small set of discrete values (e.g., 1 to 5 stars) while optimization techniques (e.g., those for solving Eq. (1)) usually work on the continuous space.

---

**Algorithm 1** The debugging algorithm for each partition

---

**Input:** debug set $\Lambda$ with adjacent matrix $A$; validation set $\Gamma$; current CF model with loss function $L$; maximum number $K$ of identified maverick ratings;
**Output:** set $\Phi$ of maverick ratings with adjacent matrix $C$;
1: initialize $\widetilde{A}^0$ as $A$;
2: initialize $\Theta^0 \leftarrow \mathrm{CF}(A)$;
3: **for** $i = 1 \rightarrow t$ **do**
4:    $\widetilde{A}^i \leftarrow \mathrm{Proj}_{\mathbb{A}}(\widetilde{A}^{i-1} - \eta \cdot \bigtriangledown_{\widetilde{A}} L_{\Gamma}(\Theta^{i-1}))$;
5:    $\Theta^i \leftarrow \mathrm{iCF}(\widetilde{A}^i, \Theta^{i-1})$;
6: **end for**
7: $C = A - \widetilde{A}^t$;
8: **return** top-$K$ candidates with largest absolute values in $C$;

---

To address the above challenges, we follow [11, 7] to first relax the ratings to continuous values, and use the *projected gradient descent* method to find the near-optimal solution for Eq. (2), i.e.,

$$\widetilde{A}^{t+1} = Proj_{\mathbb{A}}(\widetilde{A}^t - \eta \cdot \bigtriangledown_{\widetilde{A}} L_{\Gamma}), \tag{4}$$

where $\widetilde{A}$ denotes the adjacency matrix of $(\Lambda - \Phi)$ with $\widetilde{A}^0 = A$. The superscript indicates the iteration number, and $\eta$ is the learning step size. In the above equation, the projection operation $Proj_{\mathbb{A}}$ keeps $\widetilde{A}$ within the following feasible region $\mathbb{A}$, i.e., we only modify or delete the existing ratings.

$$\mathbb{A} = \{A \in \mathbb{R}^{m \times n} : a_{i,j} = 0, \forall(i,j) \notin \Lambda;\ r_{min} \leq a_{i,j} \leq r_{max}, \forall(i,j) \in \Lambda\}.$$

When $\widetilde{A}$ is obtained, we compute $C = A - \widetilde{A}$ and choose $K$ ratings in $C$ with the highest absolute values as the maverick ratings. For these maverick ratings, we either delete them or correct/modify them as indicated by the optimization results. We will empirically evaluate these two choices in the experimental section.

*(B) Computing $\bigtriangledown_{\widetilde{A}} L_{\Gamma}$.* The remaining problem of solving Eq. (2) is to compute the gradient $\bigtriangledown_{\widetilde{A}} L_{\Gamma}$ in Eq. (4). For this purpose, we first apply the chain rule as follows,

$$\bigtriangledown_{\widetilde{A}} L_{\Gamma} \quad = \quad \bigtriangledown_{\widetilde{A}} \Theta \cdot \bigtriangledown_{\Theta} L_{\Gamma} \quad = \quad \bigtriangledown_{\widetilde{A}} \widetilde{U} \cdot \bigtriangledown_{\widetilde{U}} L_{\Gamma} + \bigtriangledown_{\widetilde{A}} \widetilde{V} \cdot \bigtriangledown_{\widetilde{V}} L_{\Gamma}. \tag{5}$$

For Eq. (5), the computations of $\bigtriangledown_{\widetilde{U}} L_{\Gamma}$ and $\bigtriangledown_{\widetilde{V}} L_{\Gamma}$ are dependent upon the loss function of the CF model. For example, if we use the loss function defined in Eq. (1), we have

$$\frac{\partial L_{\Gamma}}{\partial \widetilde{u}_i} = 2 \sum_{(i,j) \in \Gamma} (\widetilde{u}_i \widetilde{v}_j^T - a_{i,j}) \widetilde{v}_j, \qquad \frac{\partial L_{\Gamma}}{\partial \widetilde{v}_j} = 2 \sum_{(i,j) \in \Gamma} (\widetilde{u}_i \widetilde{v}_j^T - a_{i,j}) \widetilde{u}_i. \tag{6}$$

As for $\bigtriangledown_{\widetilde{A}} \widetilde{U}$ and $\bigtriangledown_{\widetilde{A}} \widetilde{V}$, directly computing them would be difficult as they involve a bi-level optimization problem. Here, we resort to the KKT conditions which are applicable for many factorization-based loss functions. Due to the space limit, we directly present the resulting equations as follows and the readers may refer to [11, 7] for additional details.

$$\frac{\partial \widetilde{u}_i}{\partial a_{i,j}} = (\lambda_u I_k + \sum_{(i,j) \in \Lambda} \widetilde{v}_j^T \widetilde{v}_j)^{-1} \widetilde{v}_j^T, \qquad \frac{\partial \widetilde{v}_j}{\partial a_{i,j}} = (\lambda_v I_k + \sum_{(i,j) \in \Lambda} \widetilde{u}_i^T \widetilde{u}_i)^{-1} \widetilde{u}_i^T. \tag{7}$$

*(C) The Overall Algorithm.* The overall algorithm for solving Eq. (2) is summarized in Alg. 1, where we again ignore the regularization terms for brevity. We start with using $A$ as the initial $\widetilde{A}$, and then iteratively update $\widetilde{A}$ so as to minimize the loss of Eq. (3). In each iteration, we first compute $\widetilde{A}$ via Eq. (4), and then update the parameters via an incremental method (denoted as 'iCF'). To be specific, we use the alternating lease squares (ALS) method [4] to learn the CF parameters, and incrementally update them by running one more ALS iteration based on the parameters from the previous iteration. Finally, we choose the top-$K$ entries in $C$ with largest magnitude as the candidates of maverick data. We consider the following two possible types of editing on the spotted maverick ratings. For deleting

ratings, we directly delete the spotted maverick ratings; for modifying ratings, we add the deviations back to the original ratings (i.e., we directly use the corresponding values in $\widetilde{A}^t$ as the modified ratings).

*(D) Time Complexity.* For solving the CF model in Line 2, we use ALS [4] which takes $O((n + m)k^3 l + |\Lambda|k^2 l)$ time where $l$ is the maximum iteration number of ALS. For Line 4, it costs $O(|\Omega|k + (n + m)k^3 + |\Lambda|k^2)$ time as we only keep the values corresponding to the observed entries in $A$. For Line 5, we update the parameters based on the ones from the previous iteration, whose time complexity is $O((n + m)k^3 + |\Lambda|k^2)$. Then, the overall complexity of the algorithm is $O(|\Omega|kt + (n + m)k^3(l + t) + |\Lambda|k^2(l + t))$. By ignoring the small constants (e.g., the iteration numbers $l$ and $t$ to convergence are both around 20), the complexity reduces to $O(|\Omega| + n + m)$, which is linear w.r.t. the dataset size.

## 3.2 The Aggregation Stage

In the previous debugging stage, we obtain the maverick ratings in each partition. Next, we aggregate the results from each partition. In particular, we first perform the aggregation for each divided fold and select the candidates with largest absolute values in the $C$ matrix. Then, we keep only the entries which are always positive/negative in all $C$ matrices. That is, we filter out the values when there are opinion discrepancies (i.e., raise a rating vs. lower a rating) in different partitions. Finally, we compute the average differences of the $C$ matrices and select the entries with the largest differences as aggregated maverick ratings.

## 3.3 Debugging Wrong Prediction

The proposed method can be easily adapted to debug a wrong prediction by providing intuitive explanations. To be specific, if we use the wrong prediction as the validation set in Eq. (2), the optimization goal becomes identifying a small set of existing ratings that can maximally correct the wrong prediction. In other words, such identified ratings are most responsible for the inaccurate prediction in the first place.

## 3.4 Discussions and Generalizations

Here, we discuss the generalizations of the proposed approach.

First, note that while we study the collaborative filtering model in Eq. (1) in this paper, the proposed framework is applicable in many other collaborative filtering models as long as the following two theoretical conditions are satisfied. First, the $\triangledown_\Theta L_\Gamma$ term in Eq. (5) can be optimized to (or near) 0 (i.e., the KKT condition). Second, the second partial derivatives of $L_\Gamma$ exist so that we can use the the implicit function existence theorem [5] to obtain Eq. (7). Both conditions can be easily satisfied in many collaborative filtering models even including some neural network models, and we leave such extensions as future work.

Second, one basic assumption behind CFDEBUG is that most of the existing ratings are reliable, and a few noisy ratings in the validation set would not significantly affect the effectiveness of the proposed approach. In our experiments, we found that directly using existing ratings as the validation set works well. We also test the case when we intentionally inject some noisy data into the validation set. As we will later see, the results show that a few injected noisy ratings would not significantly affect the effectiveness of the proposed approach.

Third, the proposed approach provides a way to further improve the accuracy of future recommender systems. On one hand, the proposed approach can be directly used to debug the ratings so as to improve the overall performance. On the other hand, the proposed approach may hurt the performance on users who have many maverick ratings. To this end, we could pay special attention to these users by using different recommendation strategies.

Fourth, the proposed approach is very flexible and can be adapted for many other tasks. For example, we currently present the algorithm details for the case of explicit ratings; the proposed framework is sufficiently general to handle implicit data. Additionally, although we focus on the accuracy aspect of CF in this work, the proposed approach can be adapted to check other aspects of recommenders. We also leave these extensions as future work.

Table 1: The effectiveness results of CFDEBUG on the MovieLens data. The RMSE result of the original CF is 0.9134. The results show that the proposed CFDEBUG can significantly improve the recommendation accuracy in all cases. (We also conduct a significance testing. For all the tables, ●/∗ indicates the result is significantly better/worse than the original CF model with $p$-value$< 0.01$, and ○ indicates no significant difference.)

| Method | | eMF | NrMF | NoiseCorrection | CFDEBUG-full | CFDEBUG |
|---|---|---|---|---|---|---|
| modify ratings | 0.1% | 0.9134 ○ | 0.9137 ○ | 0.9126 ○ | 0.9052 ● | 0.9071 ● |
| | 0.2% | 0.9140 ○ | 0.9137 ○ | 0.9142 ○ | 0.9011 ● | 0.9037 ● |
| | 0.5% | 0.9148 ∗ | 0.9174 ∗ | 0.9146 ∗ | 0.8943 ● | 0.8985 ● |
| | 1% | 0.9171 ○ | 0.9180 ∗ | 0.9176 ○ | 0.8880 ● | 0.8926 ● |
| | 2% | 0.9198 ○ | 0.9218 ∗ | 0.9226 ○ | 0.8812 ● | 0.8876 ● |
| | 5% | 0.9231 ∗ | 0.9334 ∗ | 0.9251 ∗ | 0.8735 ● | 0.8810 ● |
| | 10% | 0.9246 ∗ | 0.9495 ∗ | 0.9310 ∗ | 0.8695 ● | 0.8785 ● |
| delete ratings | 0.1% | 0.9141 ○ | 0.9134 ○ | 0.9123 ○ | 0.9073 ● | 0.9073 ● |
| | 0.2% | 0.9152 ○ | 0.9134 ○ | 0.9147 ○ | 0.9033 ● | 0.9059 ● |
| | 0.5% | 0.9168 ○ | 0.9144 ○ | 0.9141 ∗ | 0.8982 ● | 0.9014 ● |
| | 1% | 0.9203 ○ | 0.9172 ∗ | 0.9165 ○ | 0.8927 ● | 0.8969 ● |
| | 2% | 0.9224 ∗ | 0.9189 ∗ | 0.9211 ○ | 0.8876 ● | 0.8954 ● |
| | 5% | 0.9295 ∗ | 0.9295 ∗ | 0.9259 ∗ | 0.8856 ● | 0.8939 ● |
| | 10% | 0.9322 ∗ | 0.9363 ∗ | 0.9312 ∗ | 0.8889 ● | 0.9036 ● |

## 4 Experimental Evaluations and Findings

### 4.1 Experimental Setup

*Datasets*. We use two publicly available benchmark datasets: *MovieLens*[2] and *Douban* [27]. The MovieLens dataset consists of 6,040 users and 3,706 items with about 1,000,000 ratings in the discrete scale of [1-5]. The Douban dataset includes 3,022 users and 6,971 items with 195,493 ratings in the discrete scale of [1-5]. For these datasets, we randomly select 80% ratings for training and use the rest 20% ratings for testing. For the training set, we split it into several folds of debug set and validation set. After the CFDEBUG finishes training on the training set as illustrated in Fig. 1, we evaluate its performance on the test set. Finally, we repeat the above process five times to do cross-validation.

*Comparison Approaches*. To show the effectiveness of the proposed approach, we compare it with the following baselines. (1) *eMF*. This approach directly applies the CF model on the existing ratings, and outputs the ones with the largest training loss as the maverick data. In this work, we use the CF model in Eq. (1) and compute $|a_{i,j} - u_i v_j^T|$ on the observed ratings to identify the maverick data. (2) *NrMF* [18]. NrMF uses a non-negative residual matrix factorization framework to detect graph anomalies. It further adds a non-negative constraint on the residual matrix. (3) *NoiseCorrection* [1]. NoiseCorrection is proposed to correct the noisy data for recommender systems. The basic idea is to cluster the users and items, and spot the outliers based on the clustering results.

*Parameters and Initializations*. In our experiments, we randomly split the training data into four folds. That is, we have four partitions and each of the four folds serves as the validation set $\Gamma$ in these partitions correspondingly. For the parameters of the CF model in the MovieLens dataset, we set $\lambda_u = 0.1$, $\lambda_v = 0.1$, and $k = 10$. For the smaller Douban dataset, we set $\lambda_u = 0.5$, $\lambda_v = 0.5$, and $k = 5$.[3]

### 4.2 Experimental Results

In our experiments, we adopt the widely-used root mean square error (RMSE) to measure the performance of the CF model.[4]

*(A) Effectiveness of* CFDEBUG. The first experiment is designed to evaluate the effectiveness of CFDEBUG. That is, when modifying/deleting some maverick ratings in the input rating matrix, to what extent the performance of the current CF model can be improved. The results are summarized

Table 2: The effectiveness results of CFDEBUG on the Douban data. The RMSE result of the original CF is 0.8751. The results show that the proposed CFDEBUG can significantly improve the recommendation accuracy in most cases.

| Method | | eMF | NrMF | NoiseCorrection | CFDEBUG-full | CFDEBUG |
|---|---|---|---|---|---|---|
| modify ratings | 0.1% | 0.8779 ○ | 0.8750 ○ | 0.8694 ● | 0.8713 ○ | 0.8714 ● |
| | 0.2% | 0.8770 ○ | 0.8747 ○ | 0.8704 ○ | 0.8673 ● | 0.8700 ● |
| | 0.5% | 0.8751 ○ | 0.8765 ○ | 0.8673 ○ | 0.8639 ● | 0.8661 ● |
| | 1% | 0.8754 ○ | 0.8760 ○ | 0.8669 ● | 0.8555 ● | 0.8621 ● |
| | 2% | 0.8776 ○ | 0.8809 * | 0.8629 ● | 0.8450 ● | 0.8505 ● |
| | 5% | 0.8835 * | 0.8862 * | 0.8615 ● | 0.8263 ● | 0.8358 ● |
| | 10% | 0.8842 * | 0.8945 * | 0.8613 ○ | 0.8117 ● | 0.8247 ● |
| delete ratings | 0.1% | 0.8779 ○ | 0.8782 ○ | 0.8708 ○ | 0.8724 ○ | 0.8731 ○ |
| | 0.2% | 0.8780 ○ | 0.8791 ○ | 0.8728 ○ | 0.8713 ○ | 0.8699 ● |
| | 0.5% | 0.8781 ○ | 0.8821 * | 0.8673 ● | 0.8655 ● | 0.8668 ○ |
| | 1% | 0.8767 ○ | 0.8836 * | 0.8669 ● | 0.8577 ● | 0.8645 ● |
| | 2% | 0.8776 ○ | 0.9013 * | 0.8682 ○ | 0.8442 ● | 0.8542 ● |
| | 5% | 0.8871 * | 0.9156 * | 0.8733 ○ | 0.8369 ● | 0.8399 ● |
| | 10% | 0.8891 * | 0.9472 * | 0.8758 ○ | 0.8324 ● | 0.8360 ● |

in Table 1 and Table 2 where we identify 0.1% - 10% ratings as maverick ones.[5] For the reported results, we retrain the model with the same parameters and initializations after the maverick ratings are modified/deleted. We also report the results (denoted as CFDEBUG-full) when we retrain CF instead of using the incremental method iCF in Line 5, Alg. 1.

From the tables, we can first observe that the proposed CFDEBUG can significantly improve the recommendation accuracy in most cases. For example, when modifying the maverick ratings on the MovieLens data and Douban data, CFDEBUG can achieve up to 3.8% and 5.7% improvements, respectively. In contrast, none of the compared methods can consistently improve the recommendation accuracy. For example, although eMF and NrMF can achieve certain improvements when the deleted/modified ratio is low, none of these improvements passes the statistical significance testing. As to NoiseCorrection, it does improve the performance in some cases on the Douban data; however, the improvement is relatively marginal. The reasons of the better performance of CFDEBUG are as follows. The compared methods spot the general data anomalies, and these anomalies do not necessarily affect the recommendation accuracy of a given CF model; in contrast, the proposed CFDEBUG directly formulates an optimization problem to minimize the prediction error on the validation set, and thus can identify the ratings that are most responsible for the degraded recommendation accuracy. Second, compared to CFDEBUG-full, CFDEBUG preserves over 98% accuracy in all the cases. The only difference between them is that CFDEBUG incrementally updates the parameters while CFDEBUG-full retrains them in Line 5, Alg. 1. In other words, the adopted incremental method is almost as accurate as the retraining method, while it is 4-5 times faster.

*(B) Effectiveness of* CFDEBUG *with Injected Noisy Data.* One basic assumption behind CFDEBUG is that most of the existing ratings are reliable. This experiment is designed to evaluate the effectiveness of CFDEBUG even when there are some noisy data in the validation sets. For this purpose, we intentionally inject some noisy ratings into the existing ones. In particular, we consider three data injection methods: *random*, *max*, and *min*, meaning randomly inject ratings with random values, 1-star ratings, and 5-star ratings, respectively. The results on the MovieLens data are shown in Table 3, where we add 1%, 2%, and 5% noisy ratings to the training data (i.e., both debug set and validation set). The 0% column is the result of the original CF (without using the proposed CFDEBUG) on the training data with injected ratings.

We can observe from Table 3 that, CFDEBUG still significantly improves the performance of the CF model even when there are intentionally injected noisy data. For example, when 2% ratings are injected, CFDEBUG can improve the original CF by up to 4.6%, 5.8%, and 5.3% under the three injection strategies, respectively. Second, we can observe that CFDEBUG can improve the performance to the level before noisy data are injected into the training data. Recall that the RMSE

Table 3: The effectiveness results of CFDEBUG when there are intentionally injected noisy ratings in the training data. The RMSE results of the original CF after adding noisy data are in the 0% columns. The results show that the proposed CFDEBUG still significantly improves the recommendation accuracy when there are injected noisy data.

| Injection | Noise | modify ratings | | | delete ratings | | |
|---|---|---|---|---|---|---|---|
| | | 0% | 1% | 10% | 0% | 1% | 10% |
| Random | 1% | 0.9163 | 0.8989 ● | 0.8788 ● | 0.9163 | 0.9002 ● | 0.9024 ● |
| | 2% | 0.9191 | 0.8963 ● | 0.8772 ● | 0.9191 | 0.8995 ● | 0.8952 ● |
| | 5% | 0.9170 | 0.9002 ● | 0.8791 ● | 0.9170 | 0.9037 ● | 0.8965 ● |
| Max | 1% | 0.9305 | 0.9044 ● | 0.8807 ● | 0.9305 | 0.9087 ● | 0.9004 ● |
| | 2% | 0.9381 | 0.9132 ● | 0.8899 ● | 0.9381 | 0.9171 ● | 0.9095 ● |
| | 5% | 0.9495 | 0.9263 ● | 0.8946 ● | 0.9495 | 0.9316 ● | 0.9153 ● |
| Min | 1% | 0.9317 | 0.9134 ● | 0.8869 ● | 0.9317 | 0.9102 ● | 0.9034 ● |
| | 2% | 0.9355 | 0.9203 ● | 0.8977 ● | 0.9355 | 0.9205 ● | 0.9090 ● |
| | 5% | 0.9469 | 0.9265 ● | 0.8964 ● | 0.9469 | 0.9288 ● | 0.9169 ● |

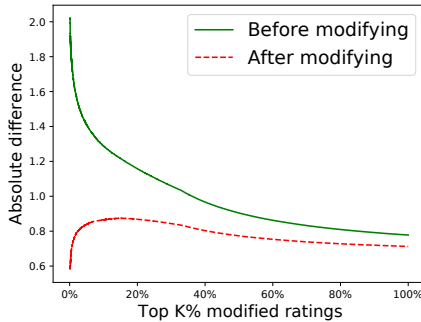

(a) Distance between the identified rating and average rating

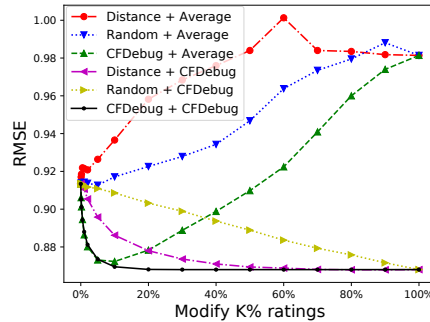

(b) Different strategies to identify and modify ratings

Figure 2: Trade-off between accuracy and personaliation. The left figure shows that the identified ratings by CFDEBUG tend to deviate from the average and CFDEBUG tends to modify it towards the average. The right figure shows that simply identifying the maverick ratings based on the personalization degree or replacing the identified ratings to average does not work as well as CFDEBUG.

result of the original CF is 0.9134 on the MovieLens data. For example, with max injection and modifying ratings, and even when 5% noisy ratings are injected (the RMSE increases to 0.9495), CFDEBUG can decrease the RMSE to 0.8946. This result is 5.8% better than the result after injection, and even 2.1% better than the result before injecting any noisy ratings.

*(C) Trade-off between Accuracy and Personalization.* Next, we study the personalization aspect of the identified ratings by CFDEBUG. To be specific, for each identified maverick rating, we compute the absolute difference between this rating and the average rating of the corresponding item, as well as the absolute difference between the modified rating by CFDEBUG and the average rating. The results of MovieLens data are shown in Fig. 2(a) where the x-axis is the percentage of debugged ratings. We observe that the top maverick ratings are more deviating from the average (the green curve), while CFDEBUG tends to modify them towards the average (the red curve).

To further check the trade-off between accuracy and personalization, we divide the proposed CFDE-BUG into two orthogonal steps, i.e., identifying the ratings and modifying the ratings. For the first step, we consider two alternative methods, i.e., *Distance* which sorts all the ratings in the descending order by their distance to the average ratings of the corresponding items, and *Random* which randomly shuffles the order of the identified maverick ratings by CFDEBUG. For the second step, we consider one alternative method *Average* which directly replaces the identified ratings with the average ratings of the corresponding items. Combining these two orthogonal steps results in six combinations as shown in Fig. 2(b). We can first observe from the figure that the proposed CFDEBUG performs best and the RMSE decreases fast when the top maverick ratings are modified. Second, in the first step,

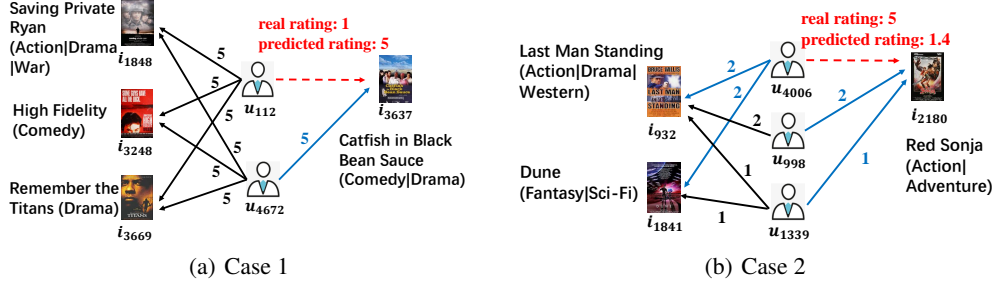

Figure 3: A case study of debugging wrong prediction by generating intuitive explanations. Movie titles and genres are also shown. The red dashed line is the inaccurate prediction to debug, and the solid lines are the corresponding explanations. For example, in the left case, "5 stars" rating from user 4672 to item 3637 (the blue solid line on the right) is most responsible for the wrong prediction (the red dashed line).

CFDEBUG is better than *Random* and *Distance*. For example, CFDEBUG can decrease the RMSE much faster than '*Random* + CFDEBUG' and '*Distance* + CFDEBUG'. This result indicates that the order of identified ratings by CFDEBUG matters, and that simply considering the most-personalized ratings is less effective in terms of improving the accuracy performance. Third, CFDEBUG is also better than *Average* in the second step, meaning that the rating modification strategy of CFDEBUG is better than simply changing ratings to average. This is consistent with the results in Fig. 2(a), where CFDEBUG changes the maverick ratings close to, but not exactly equal to the average (the red curve). Based on the debugging results, we further check the performance of users with overly personalized ratings. To be specific, we first calculate the average rating difference before debugging and after debugging for each user, and then rank the users with higher differences at higher positions. We found that the result follows the "80-20 rule", i.e., 80% users whose RMSE results become worse on the validation set are within top 22.5% users in the ranking list.

Overall, the results of this set of experiments present some important and interesting implications. On one hand, although maverick (overly personalized) ratings can help the recommender system to highlight a specific user's personal preference, they might compromise the overall accuracy. One the other hand, it is non-trivial to deal with the maverick/overly personalized ratings, since simply identifying the most distant ratings w.r.t. the average as maverick ones and/or modifying them as the average is not as effective as the proposed CFDEBUG.

*(D) Debugging Wrong Prediction*. Finally, as mentioned before, CFDEBUG can be used to provide intuitive explanations for an inaccurate prediction. Here, we present a case study on the MovieLens data in Fig. 3. Take the left case as an example. The trained CF model predicts a "5 stars" from user 112 to item 3637, while the real rating in the test set is "1 star". To debug the reason for this wrong prediction, we use it as the validation set in Eq. (2), and use the proposed CFDEBUG to identify the ratings in the training set that are most responsible for this inaccurate prediction. The results are shown in the figure. In the first example, the top-1 maverick rating identified by CFDEBUG is a "5 stars" rating from user 4672 to item 3637. By comparing the common items of users 112 and 4672, we find that they all give "5 stars" ratings to items 1848, 3248, and 3669. This result explains the plausible reason for a "5 stars" (wrong) prediction from user 112 to item 3637. Likewise, in the second example, CFDEBUG identifies several maverick ratings (denoted by blue lines) for the wrong prediction from user 4006 to item 2180. We further find that these ratings are related to two users (998 and 1339) who share similar interests to user 4006.

## 5 Conclusions

In this paper, we propose a data debugging approach to identify the maverick ratings and improve the overall recommendation accuracy for factorization-based collaborative filtering. The key idea is to search and edit the maverick ratings in order to near-optimally improve the recommendation accuracy on the validation sets. Experimental evaluations on two benchmark datasets not only demonstrate the effectiveness of the proposed approach, but also elucidate the role of maverick ratings in terms of the overall recommendation accuracy and personalization. Future directions include generalizing the proposed approach to other collaborative filtering models, and investigating the relationship between maverick ratings and the fairness of recommender systems.

## Broader Impact

In this paper, researchers introduce a data debugging method for factorization-based collaborative filtering which improves the recommendation by identifying and correcting the overly personalized ratings in recommendation systems.

As far as we know, researches on collaborative filtering have mainly focused on two directions: using advanced models and using additional information, yet few papers explore data from the overly personalized aspect. The current research suggests a new direction for collaborative filtering, orthogonal to the classical two directions. The proposed method, together with others, can improve the accuracy of recommendation systems, which will further ease the process of information acquiring. In a world locked down today due to the impact of coronavirus, easy information acquiring can give those who are not familiar with the Internet, especially disadvantaged people, many conveniences in acquiring necessities and public information online.

The current work tries to spot minorities who are identified as over-personalized and decrease their impact in terms of affecting the overall recommendation accuracy. However, it can work naturally in a reverse way: giving more priority to minorities, as a necessary step in the proposed algorithm is to identify the minorities. From the algorithm aspect, we can design special treatment for these minorities. For the social aspect, the idea of this work can be extended to much broader areas like opinion mining and decision making. For example, policymakers can understand better what kind of people are counted as minorities and how the minorities impact the final output; they may also pay special attention to minorities by giving them more weights in future decision making.

Finally, there may be a trend of making 'more personalized' recommendations. Although in the current paper, we trade personalization for accuracy, our proposed method also provides an access to those more 'personalized' data. Further research may be invoked on these personalized data, working towards satisfying both population and personalization.

## Acknowledgments and Disclosure of Funding

This work is supported by the National Natural Science Foundation of China (No. 61932021, 61690204, 61672274), and the Collaborative Innovation Center of Novel Software Technology and Industrialization. Hanghang Tong is partially supported by NSF (1947135, 2003924, and 1939725). We would like to thank Xiaofei Zhang, Zenan Li as well as the anonymous reviewers for their constructive comments. Yuan Yao is the corresponding author.

## Footnotes

[1]Hereinafter, we interchangeably use 'overly personalized ratings' and 'maverick ratings'.

[2]https://movielens.org/

[3]The code of CFDEBUG is available at https://github.com/SoftWiser-group/CFDebug.

[4]We also evaluate the ranking scenario to identify the high-rated items. The results are included in the supplementary material, which show significant improvements w.r.t. Hit Rate and nDCG metrics.

[5]The optimal ratio of maverick ratings is around 5%-10% for the studied datasets. That is, the result stabilizes when more than 10% ratings are edited.

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
