[Supplementary Material]

# Trading Personalization for Accuracy: Data Debugging in Collaborative Filtering (Supplementary Material)

**Long Chen[1], Yuan Yao[1], Feng Xu[1], Miao Xu[2,3], Hanghang Tong[4]**
[1]State Key Laboratory for Novel Software Technology, Nanjing University, China
[2]The University of Queensland, Australia     [3]RIKEN AIP, Japan
[4]University of Illinois Urbana-Champaign, USA
ronchen@smail.nju.edu.cn, {y.yao, xf}@nju.edu.cn, miao.xu@riken.jp, htong@illinois.edu

This document contains experimental details and additional experimental results for the paper "Trading Personalization for Accuracy: Data Debugging in Collaborative Filtering".

## 1   Experimental Details

*Data Splits*. In one run of the experiment, we randomly select 80% ratings for training and use the rest 20% for testing. The training data is further randomly split into four partitions, following the procedure of our proposed algorithm depicted in Fig.1.

*Hyper-parameters*. For the hyper-parameters ($\lambda_u$ and $\lambda_v$) of the CF model, we set $\lambda_u = \lambda_v$ to reduce the search space for hyper-parameters. We then search them from {0.01, 0.05, 0.1, 0.5, 1} via cross-validation. Our cross-validation results give $\lambda_u = \lambda_v = 0.1$ for MovieLens data and $\lambda_u = \lambda_v = 0.5$ for Douban data. For the hyper-parameter fold number of CFDEBUG, we find that CFDEBUG is robust against this parameter. We will show the experimental results later for different values of fold number. For the results reported in the paper, we set it to 4. An additional hyper-parameter for both CF and CFDEBUG is the rank $k$. Based on previous literature, a relatively small rank is usually enough for effectively reconstructing the rating matrix [4, 1]. Therefore, we set $k = 10$ for the larger MovieLens data and $k = 5$ for the smaller Douban data. In this document, we will also provide sensitivity experiments about this parameter.

*Evaluation Metrics*. We use RMSE to evaluate the rating prediction performance of the CF model in the main paper. RMSE is formally defined as

$$RMSE = \sqrt{\frac{\sum_{(i,j) \in D_T} \left( \hat{r}_{i,j} - r_{i,j} \right)^2}{|D_T|}}, \tag{8}$$

where $D_T$ denotes the test set, $\hat{r}_{i,j}$ is the rating from user $i$ to item $j$ predicted by the CF model, and $r_{i,j}$ is the real rating in the test set.

In addition to rating prediction scenario, the ranking scenario is also widely studied in evaluating the performance of CF models. Following [2, 3], we select all the high ratings (4 and 5 stars) in the test set and accompany each of them with randomly selected 100 unrated items from the same user as negative samples. We then output a ranked list of each user based the predicted rating. We use HR (Hit Rate) and nDCG (normalized Discounted Cumulative Gain) to evaluate the ranking performance. The HR and nDCG are defined as follows.

$$HR@K = \frac{1}{|D_T|} \sum_{t=1}^{|D_T|} \text{hit}_t, \tag{9}$$

Table 4: The standard deviation results of CFDEBUG on the MovieLens data (the mean results are in Table 1 of the main paper).

| Method | | eMF | NrMF | NoiseCorrection | CFDEBUG-full | CFDEBUG |
|---|---|---|---|---|---|---|
| modify ratings | 0.1% | 0.0020 | 0.0025 | 0.0023 | 0.0012 | 0.0010 |
| | 0.2% | 0.0013 | 0.0025 | 0.0022 | 0.0012 | 0.0008 |
| | 0.5% | 0.0021 | 0.0018 | 0.0024 | 0.0008 | 0.0010 |
| | 1% | 0.0015 | 0.0019 | 0.0034 | 0.0009 | 0.0010 |
| | 2% | 0.0017 | 0.0025 | 0.0030 | 0.0009 | 0.0010 |
| | 5% | 0.0028 | 0.0021 | 0.0022 | 0.0006 | 0.0003 |
| | 10% | 0.0021 | 0.0103 | 0.0017 | 0.0006 | 0.0004 |
| delete ratings | 0.1% | 0.0020 | 0.0024 | 0.0020 | 0.0011 | 0.0007 |
| | 0.2% | 0.0010 | 0.0025 | 0.0024 | 0.0014 | 0.0016 |
| | 0.5% | 0.0018 | 0.0026 | 0.0025 | 0.0019 | 0.0021 |
| | 1% | 0.0013 | 0.0018 | 0.0028 | 0.0013 | 0.0009 |
| | 2% | 0.0017 | 0.0043 | 0.0036 | 0.0010 | 0.0011 |
| | 5% | 0.0035 | 0.0035 | 0.0032 | 0.0020 | 0.0018 |
| | 10% | 0.0018 | 0.0026 | 0.0020 | 0.0010 | 0.0022 |

Table 5: The standard deviation results of CFDEBUG on the Douban data (the mean results are in Table 2 of the main paper).

| Method | | eMF | NrMF | NoiseCorrection | CFDEBUG-full | CFDEBUG |
|---|---|---|---|---|---|---|
| modify ratings | 0.1% | 0.0022 | 0.0004 | 0.0018 | 0.0018 | 0.0015 |
| | 0.2% | 0.0014 | 0.0004 | 0.0032 | 0.0012 | 0.0022 |
| | 0.5% | 0.0005 | 0.0006 | 0.0034 | 0.0022 | 0.0025 |
| | 1% | 0.0009 | 0.0014 | 0.0052 | 0.0018 | 0.0024 |
| | 2% | 0.0015 | 0.0011 | 0.0052 | 0.0016 | 0.0024 |
| | 5% | 0.0023 | 0.0023 | 0.0033 | 0.0020 | 0.0028 |
| | 10% | 0.0018 | 0.0042 | 0.0045 | 0.0025 | 0.0022 |
| delete ratings | 0.1% | 0.0023 | 0.0008 | 0.0021 | 0.0024 | 0.0014 |
| | 0.2% | 0.0021 | 0.0007 | 0.0034 | 0.0013 | 0.0031 |
| | 0.5% | 0.0010 | 0.0022 | 0.0033 | 0.0024 | 0.0013 |
| | 1% | 0.0013 | 0.0041 | 0.0035 | 0.0027 | 0.0041 |
| | 2% | 0.0030 | 0.0025 | 0.0026 | 0.0032 | 0.0042 |
| | 5% | 0.0028 | 0.0047 | 0.0051 | 0.0026 | 0.0028 |
| | 10% | 0.0021 | 0.0061 | 0.0026 | 0.0029 | 0.0026 |

where $D_T$ is the test set, and $hit_t$ equals to 1 when the corresponding test rating is within top-$K$ positions in the ranked list.

$$nDCG@K = \frac{1}{|D_T|} \sum_{t=1}^{|D_T|} \frac{\log 2}{\log (r_t + 1)}, \tag{10}$$

where $r_t$ is the ranking position in the ranked list of a test rating. $r_t$ is infinite if the ranking position of the test rating is not within top-$K$.

*Implementations.* The proposed approach is implemented with Python, and all the experiments are run on a desktop computer with 6 CPU cores at 2.6G Hz.

## 2 Additional Experimental Results

*Standard Deviation Results of Multiple Runs.* In addition to the mean results we reported in the main paper, here we report the standard deviation results of 5 random experiments. The results are shown in Table 4 and Table 5.

*Parameter Sensitivity.* Next, we investigate the performance of CFDEBUG w.r.t. two parameters. The first one is the number of folds/partitions to divide the training data, and the second one is the the latent rank $k$ from the original CF model. We only present the results on MovieLens for brevity and

| (a) Fold number | (b) Rank $k$ |
|---|---|

Figure 4: Parameter sensitivity results. The proposed CFDEBUG can significantly improve the recommendation accuracy when we vary these parameters in a relatively wide range.

Table 6: Performance of ranking metrics. The proposed CFDEBUG can also yield significant improvements.

|  | Original CF | 0.1% | 0.2% | 0.5% | 1% | 2% | 5% | 10% |
|---|---|---|---|---|---|---|---|---|
| HR@5 | 0.11 | 0.12 | 0.13 | 0.14 | 0.16 | 0.18 | 0.20 | 0.21 |
| HR@10 | 0.28 | 0.29 | 0.30 | 0.31 | 0.33 | 0.35 | 0.37 | 0.38 |
| HR@20 | 0.53 | 0.54 | 0.55 | 0.56 | 0.57 | 0.58 | 0.59 | 0.60 |
| nDCG@5 | 0.05 | 0.06 | 0.06 | 0.07 | 0.08 | 0.10 | 0.11 | 0.12 |
| nDCG@10 | 0.11 | 0.11 | 0.12 | 0.13 | 0.14 | 0.15 | 0.17 | 0.18 |
| nDCG@20 | 0.17 | 0.18 | 0.18 | 0.19 | 0.20 | 0.21 | 0.22 | 0.23 |

similar results are observed on Douban. As we can see from Fig. 4, overall, CFDEBUG is robust w.r.t. to the two parameters in a relatively wide range. For example, when we change the fold number from 3 to 6, or when we vary the latent rank from 10 to 50, CFDEBUG consistently achieves significant improvements compared to the original CF model.

*Performance in Ranking Scenario.* For the ranking performance, we show the results on MovieLens in Table 6. The first column is the metric. The second column contains the results of the original CF. The rest columns are the results of the proposed CFDEBUG when modifying different data percentages (0.1% - 10%). We can see that the proposed method can still improve the recommendation accuracy in terms of the ranking metrics (e.g., $0.11 \rightarrow 0.21$ on the HR@5 metric). Overall, this experiment shows that the RMSE improvements of the proposed method can also yield significant improvements in the ranking metrics.

*Running Time.* One advantage of our proposed CFDEBUG is that it can do parallel computation. Here we create $K$ processes if we divide the training data into $K$ partitions, and each process does the computation of each partition. For MovieLens data, CFDEBUG takes 20 seconds for each iteration in the debugging stage and CFDEBUG-full takes 90 seconds. For Douban data, CFDEBUG takes 10 seconds for one iteration in the debugging stage and CFDEBUG-full takes 70 seconds. It needs around 50 and 30 iterations for them to converge, respectively. Overall, CFDEBUG runs 4-5 times faster than CFDEBUG-full.