[Reviews · NeurIPS 2020]

Review 1

Summary and Contributions: This paper proposed CFDebug, an iterative approach to search for ratings without which the predictive performance could be better. Essentially CFDebug iterate between (1) performing matrix-factorization-based CF with a given rating matrix and (2) searching for maverick ratings using projected gradient descent. The approach is simple and straightforward. Though I like the idea, there are major concerns on directly overfitting the validation set, rendering the claim on accuracy invalid (see the correctness section for details).

Strengths: The perspective of data debugging for CF-based recommender systems is interesting. The proposed method is straightforward and makes sense. Interesting experiments on the trade-off between accuracy and personalization.

Weaknesses: Major concern on directly overfitting the validation set, rendering all results (on accuracy) invalid. Since neither of the MF part and the projected gradient descent part is new, there seem to be litter technical merit in CFDebug.

Correctness: One major concern that I have is on overfitting the validation set, since CFD directly update the rating matrix using the *validation* set. In this setting, the results are hardly valid, if not fraud. At the very least, CFD should include a test set (different from the training and validation set) which one can evaluate the performance on. Otherwise it is very likely that CFD may directly overfit the validation set.

Clarity: Presentation is clear and the paper is well organized.

Relation to Prior Work: Yes.

Reproducibility: Yes

Additional Feedback: It would help to explain more on what the adjacency matrix C is for in Section 2, since this is not typical in CF. In Section 4 (D), it would have been more interesting and meaningful if the movie titles can be included for the case study. ------------- AFTER REBUTTAL I have read the author feedback as well as other reviews and I thank the authors for clarifying the training/validation/test split. I would like to upgrade my rating and suggest the authors make it clearer in the final version, since the original statement “we randomly select 80% ratings for training and use the rest 20% ratings for testing” is too vague.


Review 2

Summary and Contributions: This paper proposes a data debugging framework to identify too personalized or noisy ratings which degrade the overall performance of a given recommender model. By editing noisy user ratings, it shows the performance gain in the validation set. Experimental results show that the proposed debugging framework is effective for improving the overall performance by balancing accuracy and personalization.

Strengths: 1. The idea of the paper is interesting, and the author shows the effectiveness of the proposed model using a numerical solution. 2. This paper clarifies the key novelty of the proposed model from the existing noisy correction method and outlier detection techniques. 3. It shows an extensive experimental evaluation and various ablation studies.

Weaknesses: 1. Although this paper shows the improvement of the overall accuracy, it is also interesting to show whether the accuracy of overly personalized users, i.e., maverick users, decreases or not. 2. Since the proposed framework seems to be universal, it would be interesting for applying the proposed framework to various baseline models, including PMF, SVD++, and AutoRec. Please refer to the papers. Suvash Sedhain et al., "AutoRec: Autoencoders Meet Collaborative Filtering," WWW 2015 3. The datasets. e.g., ML-1M and Douban are rather small. It would be better to evaluate the proposed framework with large-scale datasets such as ML-10M, ML-20M, and Netflix. 4. Although the top-N recommendation metrics are discussed in the Appendix, the top-N recommendation problem would be more interesting than the rating prediction problem. It would be better to show some experimental results for the top-N recommendation in the main pages.

Correctness: The proposed method is correct. However, it would be better to show a more thorough evaluation of large-scale datasets and various baseline models. - Minor typo Page 8: One one hand, although maverick ratings can -> On one hand, although maverick ratings can

Clarity: This paper is well-written, and the problem is interesting.

Relation to Prior Work: It would be welcome to compare the proposed model with the existing noisy correction/detection methods in recommender models, as cited in [7]. - Dongsheng Li et al., "Collaborative Filtering with Noisy Ratings,” SDM 2019

Reproducibility: Yes

Additional Feedback: After rebuttal, I would recommend that this paper shows a more detailed analysis of performance gain in the final draft. As mentioned in the rebuttal, while some users get performance gains, other users are worse performance gains as the trade-off. Also, I would like to show an extension of the proposed model for the top-N recommendation setting.


Review 3

Summary and Contributions: This paper focuses on the matrix factorization model for collaborative filtering problem, and proposes an algorithm to identify observed ratings in user-item rating matrix that could hurt model's generalization performance. The main idea is to backpropagate validation loss to update observed ratings in the training set, and then modify or delete training labels that belong to largest-updated entries. They adopt a cross-validation and aggregation paradigm to find target ratings in the whole training data. Experimental results demonstrate the effectiveness of the proposed approach.

Strengths: S1. The formulated problem in this paper is novel and interesting. Different from existing works on anomaly detection or robust learning for collaborative filtering, this paper aims to find out and modify harmful ratings in observed entries to improve model's generalization, which is from a new perspective. S2. The paper is generally well-organized and well-written. The motivation and problem formulations are clearly stated. The algorithm is adequately described and easy to understand. S3. The experimental results are promising, which significantly outperforms all the compared methods. S4. The topic of the paper is relevant to the NeurIPS community.

Weaknesses: W1. The proposed algorithm is limited to matrix factorization model, and can be hardly extended to more state-of-art neural network-based latent factor models proposed in recent years. Because the derivate of model parameters with respect to training labels in Equation (7) needs to be a closed form solution as in matrix factorization. This may restrict a broader impact of the proposed solution. W2. I'm concerned about the authors' claim on the trade-off between personalization and accuracy. As emphasized in the title, the authors consider the performance gain of the proposed algorithm as trading personalization for accuracy, but there is no direct empirical evaluation evidence to support this claim. For example, after modifying several ratings of a specific user in the training set, how does the validation performance change for this user and other users, respectively? If the validation performance decreases for the target user and increases for other users, then it demonstrates that the modified ratings are somehow ``overly personalized``. But if the validation performance increases for the target user, then it doesn't show a trade-off between personalization and accuracy. The authors need to provide related experimental results.

Correctness: The claims, method, and empirical methodology are basically correct.

Clarity: Yes, this paper is clearly structured and well-written.

Relation to Prior Work: Yes, the authors have explained how the proposed work differs from prior work in the literature.

Reproducibility: Yes

Additional Feedback: 1. See W1 and W2. 2. A question is about the main results in Table 1 and 2. It seems that the performance of CFDEBUG keeps growing with the ratio of modified or deleted ratings. What is the optimal ratio empirically? 3. The references should be consistent in format. Page numbers are provided in some references while missing in others. ----------------------------------- AFTER REBUTTAL After reading the author response and other reviews, I agree that this paper should be accepted after including experimental results about the tradeoff between personalization and accuracy in the final version. So I would like to keep my rating.


Review 4

Summary and Contributions: This paper works on the problem to reduce the impact of over-personalized ratings on the overall performance of recommendation systems. Such a problem is different from classical studies on noisy data or anomaly detection, since these methods exploiting the inconsistency, while under the current scenario, there is no inconsistency. To solve the problem, the paper proposed a data debugging method based on matrix factorization. By “data debugging”, they mean mimicking the process of code debugging, i.e., knowing where is incorrect by testing. Thus, they divided the data into several groups and used the validation set to test which subgroup of ratings would do the largest hurt to the overall performance. Due to the difficulty of combinatorial optimization, the paper proposed to use the difference after updating as a “measurement”. A bunch of experiments validated the effectiveness of the proposed methods from various aspects.

Strengths: The problem studied in this paper is quite novel, as few papers studied the impact of “overly-personalized” ratings. To solve the problem, the paper proposed to use a validation set to identify the subgroup of ratings that would contribute most to harm the overall performance. Such kind of method is reasonable under the implicit assumption that the noise rate is not high. The experimental results are impressive. The paper has done a bunch of experiments to show various aspects of the proposal. The experimental results on “injected noisy data” shows that even if the validation set took some “over-personalized” ratings, the performance is still good. The paper is well-organized and well-written.

Weaknesses: Besides specifying when the proposed method work, the situation under which the proposed method could fail need also to be discussed. I would expect that if a large group of ratings is overly-rated, then the method cannot work well.

Correctness: Yes, they are correct.

Clarity: Yes, this paper is well-organized and well-written.

Relation to Prior Work: Yes, it is clearly discussed how this work differs from previous contributions.

Reproducibility: Yes

Additional Feedback: Although the method works empirically well, such an assumption should be made implicit. Due to the difficulty of discrete optimization, the paper relaxed the problem to be a continuous one. This solution is novel. I am thinking of an alternative way to solve the problem from the matrix completion point of view, which has the potential to have a theoretical guarantee. Any discussion on this part? There is one suggestion for the paper: besides specifying when the proposed method work, the situation under which the proposed method could fail need also to be discussed. I would expect that if a large group of ratings is overly-rated, then the method cannot work well. Discussion on the weak part of the proposed method would make the paper stronger. ------------------------------- After Rebuttal I have read comments from other reviewers and the author's feedback. I like the idea of this work, and all claims are supported by the experiments well. I would like to upgrade my rating as well since this is a really good paper that should be presented in NeurIPS'20.

[Author Response · NeurIPS 2020]

We would like to thank reviewers for their time and constructive comments. Following are our responses to the raised questions. We start with common questions followed by individual ones.

**Common Q1**: Both **R2** and **R3** question about the claim of the tradeoff between personalization and accuracy. We empirically validated the existence of such tradeoff. For example, by tracking the performance of users in the validation set, we find that the results follow the "80-20 rule", i.e., 80% users whose RMSE results become worse are within top 22.5% over-personalized users. Following reviewers' suggestion, we will include this result into our revised version.

**Common Q2**: Both **R2** and **R3** also question about the applicability of the proposed method. Indeed, we agree with **R2** that the proposed framework is quite generic, and we would like to clarify that the proposed method does not necessarily require a closed form solution such as in matrix factorization. In fact, the proposed framework can be used in many collaborative filtering models as long as the following two theoretical conditions are satisfied. First, the $\bigtriangledown_\Theta L_\Gamma$ term in Eq. (5) can be optimized to (or near) 0 (i.e., the KKT condition). Second, the second partial derivatives of $L_\Gamma$ exist so that we can use the the implicit function existence theorem to obtain Eq. (7). Both conditions can be easily satisfied in many collaborative filtering models including the neural network models mentioned by the reviewers. We plan to extend our idea to such models in our future work.

**Q1 from R1**: **R1** has a concern about overfitting the validation set and the inclusion of a test set. We would like to clarify that this is a misunderstanding and we do have a test set which is separated from both the training set and the validation set. To be specific, as stated in Section 4.1, "we randomly select 80% ratings for training and use the rest 20% ratings for testing". For the training set, we then split it into several folds of debug set and validation set. After the CFDebug finishes training on the training set as illustrated in Fig. 1, we evaluate its performance on the test set. Finally, we repeat the above process five times to do cross-validation. We will make it clearer in the revised version.

**Q2 from R1**: **R1** also concerns about the technical merit of the proposed method. We would like to point out that the key contributions of this work are two fold. First, the data debugging framework that is potentially applicable for a large set of collaborative filtering models (see the response for **Common Q2**). Second, experimental findings and analysis about the tradeoff between personalization and accuracy (e.g., modifying the over-personalized ratings would help improve the overall accuracy).

**Other questions from R1**: Thanks for the advice, we will add more explanations about the C matrix. In Section 4 (D), we did display the movie titles in the case study. We will make it clearer in the revised version.

**Q1 from R2**: **R2** encourages us to experiment on larger datasets. Thank you for the suggestion. We have done experiments on ML-10M. The results also show the effectiveness of the proposed method. For example, with 10% modified ratings, our method gives 2.1% RMSE improvement compared to the original performance (from 0.8336 to 0.8162). We will include these results into our future version.

**Other questions from R2**: As suggested by the reviewer, we will include the top-N studies in the main paper instead of the appendix, and will consider to include the comparison with [7] in our journal version.

**Q1 from R3**: **R3** also asks about the optimal ratio of the results in Table 1 and 2. Empirically, the optimal ratio is around 5%-10% for the studied datasets. We will make this clearer in our revised version.

**Q1 from R4**. **R4** encourages us to analyze the limitation of the proposed method. Thanks for pointing this out. For example, when the ratio of "over-personalized" ratings increases to a certain extent, the overall performance might start to suffer. Much more theoretic work is needed to understand or identify such a theoretic 'transition' point.

**Q2 from R4**: **R4** also suggests using matrix completion view to provide theoretical guarantees. We agree that matrix completion has the potential to provide additional theoretical guarantee.

[Meta-Review · NeurIPS 2020]

The paper received overall very positive scores (after communication through author response). All the reviewers agree that the paper made a very interesting contribution from a novel angle to understand the tradeoff between "over-personalization" and accuracy. The empirical results provide convincing support for the claim. Therefore, I recommend acceptance. I suggest the authors incorporate the feedback from the reviewers in the revision.